# Agricultural Food System Transformation on China’s Food Security

**DOI:** 10.3390/foods12152906

**Published:** 2023-07-31

**Authors:** Sicheng Zhao, Tingyu Li, Guogang Wang

**Affiliations:** Institute of Agricultural Economics and Development, Chinese Academy of Agricultural Sciences, Beijing 100081, China

**Keywords:** food security, agri-food system transformation, coupling degree

## Abstract

Different countries and regions in the world are experiencing structural transformation of the agri-food system, which is represented by the increase of meat and feed consumption. Based on a detailed review of the global and China’s agri-food system transformation, this paper constructs an analytical framework of the impact of agri-food system transformation on food security and discusses the food security issues in China after 2000. The results show that the proportion of the dimension of agri-food system transformation in the food security index has a significant downward trend, and its positive effect on food security is decreasing. At the same time, due to the growth in demand for meat during the transformation of the agri-food system, China imports a large number of feed grains such as soybean and corn, leading to a decline in domestic food self-sufficiency. Furthermore, the coupling coordination degree between China’s agri-food system transformation and food quantity security decreases, and their development tends to deviate. In the future, increasing the consumption of grain-saving poultry and curbing table waste are feasible strategies for China to cope with the transformation of agri-food system and ensure food security. China’s problems and solutions in facing the transformation of its agri-food system can provide some references for other developing countries.

## 1. Introduction

Since 1990, the international grain output has grown at a rapid rate of 2.6 percent. In 2021, the global grain output was about 2.8 billion tons, and the disposable per capita grain was about 305 kg. At the same time, however, 193 million people in 53 countries experienced a food crisis, an increase of about 26% over 2020 [1]. On the one hand, this is influenced by exogenous factors as COVID-19, extreme climate disasters, and other factors in agricultural production [2]. On the other hand, this is also due to endogenous economic structural problems, such as growing grain consumption for biomass energy and increasing grain consumption for meat production. It is estimated that, in 2021, the global biofuel production will reach 151.3 billion litters (119.4 million tons) [3], which consuming approximately 198 million tons of grain [4]. Meanwhile, the global demand for feed grain is about 10.05 billion tons [4]. It is suggested that energy and meat production consume more than 40 percent of the world’s total grain production.

Global meat consumption has been on the rise for the past 50 years. Specifically, global meat consumption increased from 287 million to 324 million tons in the decade 2010–2019, an increase of about 12.8 percent [5]. In terms of total consumption in 2019, East Asia, the Americas, and Europe have the highest level of meat consumption, reaching 100 million tons, 90 million tons, and 55 million tons, respectively [6]. These three regions account for the vast majority of global meat consumption. In terms of per capita consumption, the per capita meat consumption of the United States, Australia, and Argentina all exceeded 100 kg [6], ranking as the top three countries in the world in 2019. In addition, total and per capita meat consumption in Africa and South Asia is lower than the world’s average, and the local agri-food system lacks resilience. It is difficult to effectively guarantee the nutrition and health of the people in Africa and South Asia, and some people have been on the brink of hunger for a long time.

Agri-food system is an extensive concept that includes all the elements and exchange relations of agriculture and food value chain. The transformation of agri-food system in this paper mainly refers to the increase of feed grain demand caused by rising meat consumption. China, as the world’s largest developing country, is also experiencing this change. Our paper aims to study the impact of the agri-food system transformation on China’s food security. This research will not only provide a new perspective for understanding China’s food security, but will also provide some lessons for other countries.

## 2. Literature Review

The research on food security has a long history and is still receiving high attention at the international scope. With the development of economy, the definition of food security has undergone a series of changes. The connotation of food security has undergone changes from macro total amount to micro individual, from production to livelihood, and from objective index to subjective perception [7]. Food security is associated with poverty, and food diversity [8], cultural acceptance [9], malnutrition and child mortality [10], environmental and climate impacts [11,12], agricultural land use structure [13], energy and water resource constraints [14,15], and other influencing factors have been gradually taken into consideration as influences. Food security has developed into a comprehensive concept including food supply, food production, fairness in food distribution, natural environment, nutrition and health, and economic and social development status.

Recently, the impact of the agri-food system transformation on food security has drawn widespread attention. The agri-food system is a generalized food system, that includes the R&D, production, circulation, consumption, nutrition, resources, environment, and other links of agricultural value chain. It also includes the economic, social and ecological results generated by the operation of the system as well as the policy, legal system and social and cultural environment that maintain the its operation [1,2]. The adjustment of residents’ food consumption structure is the driving force of the agri-food system transformation, and the increase of meat consumption is one of the most important trends of the transformation in the past 20 years. Both developed countries and areas, such as the United States, the European Union, and Australia, and emerging economies, such as China, Brazil, and India, experienced significant growth in meat consumption demand after 2000 [4]. Feed grain, as a raw material for meat production, is also an important part of the agri-food system. As mentioned above, the growth of feed grain consumption has increased the demand for grain ration, which has seriously endangered global food security. Therefore, special attention needs to be paid to the impact of the transformation of the agri-food system, marked by the growth in demand for meat, on food security. Moreover, food fraud is gradually becoming an important issue in food security. In the dairy industry, despite the continuous progress of detection technology, fraud techniques have become more refined in some cases. In the foreseeable future, food fraud remains a significant threat to resident nutrition and health [16,17].

The core issue of food security research varies for different countries. For developing countries, urban agriculture development and urban poverty reduction [18], biotechnology development [19], and the revenue driving effect of horticultural exports [20] can improve the food security situation. Conversely, the decline in R&D and infrastructure investment [21] has a negative impact on the food security situation in developing countries. Compared with developing countries, developed countries have more adequate food supply. Even if developed countries suffer from lower food self-sufficiency, they can also meet their needs through the international market. However, the fluctuation of international food prices should not be underestimated. High food prices will also put pressure on developed countries with high food dependence on foreign countries. Recently, the food self-sufficiency rate of the UK has been declining, and the food imports have been increasing. However, the impact of the decline of its own self-sufficiency on British food security is not as serious as the political and economic situation as the main source countries of food imports of the UK is stable [22,23].

China has long since had a tight balance between grain supply and demand. Ensuring a high self-sufficiency rate for grain has always been at the core of China’s grain production, and it also plays a pivotal role in world food security. The growth of feed grain production cannot meet the rapid growth of meat production in China, which eventually results in the import of large quantities of feed raw materials [24]. The growth of meat consumption has put great pressure on China’s grain self-sufficiency. Adequate attention should also be paid to the impact of the transformation of China’s agri-food system on food security.

With the increase of meat consumption, the grain demand structure of China ushered in a significant adjustment. From 2000 to 2021, China’s grain demand increased from 430 million tons to 820 million tons, an increase of 94 percent. In terms of grain demand structure, the proportion of grain ration consumption decreased gradually, and the proportion of feed grain consumption increased rapidly. The proportion of residents’ grain ration demand decreased from 44% in 2000 to 35% in 2021, while the proportion of feed grain demand increased from 20% to 33% in the same period. The consumption of grain ration gradually decreased and was replaced by the consumption of feed grain. From the perspective of subdivided varieties, the phenomenon of demand structural differentiation is still obvious, and different varieties have their own emphasis. Corn crop has been mainly used for feed consumption, and grain ration consumption accounts for only a small proportion. Wheat and rice are mainly consumed by residents, but the proportion of residents’ grain ration consumption has gradually decreased, and feeding consumption has continuously increased. From 2000 to 2021, the proportion of rice feeding consumption increased from 4% to 10%, and the proportion of wheat feeding consumption increased from 4% to 21%. The adjustment of household food consumption structure directly reflects the change of grain demand structure.

Figure 1 illustrates the logical relationship between the transformation of the agri-food system and food security in the context of the adjustment of Chinese residents’ food structure. Regarding the international situation, the outbreak of COVID-19, frequent extreme weather events, and other factors have combined to increase the international food prices, causing the China’s agri-food system to face more risk and uncertainty.

Within China, as mentioned above, the increase of household income leads to the adjustment of the food consumption structure, with a decrease in grain ration consumption and an increase in meat consumption. As one of the main food raw materials, the adjustment of food consumption structure is bound to be transmitted to the food market through the adjustment of input factors, which is reflected in the gradual decline in the proportion of grain ration consumption and the increase in feed grain consumption year by year. From the perspective of grain supply, the adjustment in grain production structure lags behind the adjustment of demand structure. Although the feed grain output increases significantly, it lags behind the growth of demand, and the import of feed grain obviously increases. Specifically, from 2000 to 2021, the domestic corn output increased from 110 million tons to 272.5 million tons, while the international trade changed from a net export to net import of 28.35 million tons. In the same period, the soybean output only increased from 15.41 million tons to 16.4 million tons, while the import volume increased from 10.42 million tons to 96.52 million tons. Feed grain import has become a key issue to ensure a higher self-sufficiency rate of grain in China.

To sum up, there are numerous studies about food security, but there are few studies about China’s food security from the perspective of agri-food system transformation. Analyzing the impact of the transformation of China’s agri-food system driven by the increase of meat demand on food security is a strong supplement to existing studies; this is the main innovation of this article. 

## 3. Methodology

Currently, there are already several widespread FSIs (food security index), such as the FAO FSI [25] and the Economist FSI [26]. In this study, the establishment of a new set of food security indices is not intend to measure food security from another perspective. The purpose of establishing an FSI in this article is to study the impact of agri-food system transformation on food security. In short, calculating an FSI is just a means rather than an end. The following are the indicator system and the food security index calculation method.

Firstly, a multi-dimensional food security index system was established, and the transformation dimension of agri-food system was included to measure China’s food security index. Secondly, on the basis of calculating the scores of different subsystems, the proportion of different subsystems in the overall index was analyzed to judge their contribution to food security. Finally, the coupling degree of different subsystems was calculated, especially the coupling coordination relationship between the dimension of agri-food system transformation and the dimension of food quantity security. Also, the impact of agri-food system transformation on food security was analyzed. The paper mainly uses annual data from China at the national level, covering the period 2000–2019.

### 3.1. Food Security Indicator System

In this paper, food security mainly includes five elements: food quantity security; food quality security; food input factor security; natural disaster impact; and agri-food system transformation. Among these, the agri-food system transformation is the core issue of this research. According to the literature review, the nutrition and health status of residents, household income and power consumption, political stability, and other factors are also important factors for country’s food security, but as they do not address the research aim of this paper, they will not be included in the subsequent food security indicator system.

According to the above definition, the evaluation index system of food security index includes five main aspects, namely quantity safety dimension, quality safety dimension, natural disaster dimension, factor input dimension, and agri-food system transformation dimension (Table 1). 

The quantity security dimension mainly includes two elements, production and grain self-sufficiency. This dimension represents the grain supply capacity and reflects whether the quantity of grain is sufficient. The quality and safety dimension only includes the food quality and safety situation. The natural disaster dimension includes two aspects: the degree of disaster and disaster management. This dimension is used to measure the damage degree of natural disasters to agricultural production and the ability to cope with natural disasters. The factor input dimension includes water resources utilization, labor input, capital input, land input, energy consumption, and other items, and mainly considers the influence of various production factors on food security. The transformation dimension of the agri-food system mainly includes the transformation of food structure, transformation of grain production and consumption structure, and the grain import situation. To sum up, the food production security indicator system in this paper contains 5 first-level indicators, 13 second-level indicators, and 30 third-level indicators.

### 3.2. Index Measurement Method

In this section, the entropy method is used to calculate the food production security index. Entropy is a measure of uncertainty. The increase in information can reduce uncertainty and, thus, the entropy value; a decrease in information will cause the entropy value to increase. The essence of measuring the food production security index is assigning different weights to different indicators and then combining the index system into a security index according to these weights. The entropy method can determine a relatively objective weight for each index by calculating the information entropy of the system and the index, thus avoiding the error caused by subjectively assigning weights to different indices. Therefore, the entropy method is commonly used to construct an index evaluation system. The specific calculation process of the food production security index is as follows.

In contrast to regression analysis, the entropy method eliminates the dimensional difference between indicators through a series of operations before weight calculation. The commonly used dimensionless methods include the standardization method, extreme value method, and vector specification method. The extreme value treatment method has the advantages of monotonicity, difference invariance, and translation independence. Therefore, this method is adopted in the construction of the index system here.

First, the indicators are divided into two categories: positive and negative. Positive indicators are those that increase the value of the food production security index, while negative indicators are those that decrease this value. In the index system constructed here, the negative indicators are as follows: (1) ratio of grain loss, (2) concentration degree of the grain import market, (3) disaster area caused by floods, (4) disaster area caused by drought, (5) proportion of grain imports, (6) adjustment degree of grain ration imports, (7) proportion of pork consumption and (8) proportion of poultry consumption. while the other indicators are all positive.

If the indicator is positive, then
(1)$$y_{ij}{}^{*}=\frac{y_{ij}-\min(y_{j})}{\max(y_{j})-\min(y_{j})}$$

If the indicator is negative, then
(2)$$y_{ij}{}^{*}=\frac{\max(y_{j}){-y}_{ij}}{\max(y_{j})-\min(y_{j})}$$

After the index is standardized, the index value is transformed into a number between 0 and 1, and the order of magnitude gap between the indicators is greatly reduced. In addition, the variable subscript *i* in Equations (1) and (2) represents the year and *j* represents different indicators; these meanings are the same in subsequent statements.

The weight of the index of a certain year in all years of the index is calculated:(3)$$p_{ij}=\frac{y_{ij}^{*}}{\sum_{i}y_{ij}^{*}}$$

The redundancy of information entropy is calculated. 

First, the information entropy of different indicators is
(4)$$s_{j}=-1/\ln(n)\sum_{i=1}^{n}p_{ij}*\ln(p_{ij})$$

In the above Equation, *n* represents the total number of years, as below.

Second, the information entropy redundancy is
(5)$$r_{j}=1-s_{j}$$

The index weight is then determined:(6)$$w_{j}=\frac{r_{j}}{\sum_{j}r_{j}}$$

The food production security index and subdimension score are generated:(7)$$index_{i}=\sum_{j=1}^{m}y_{ij}*w_{j}$$

The above formula represents the measurement method of the food production security index in year *i*, where the letter *m* represents the total number of indicators in the index system.
(8)$$score_{i}=\sum_{j=1}^{m_{k}}y_{ij}*w_{j}$$

The above formula represents the score of a subdimension in year *i*, and *m_k_* represents the number of indicators in a subdimension. The above formula shows that the food production security index is the weighted average of all indices, the subdimension score is the weighted average of the indicators, and the food production security index is the direct sum result of all subdimension indices. These basic quantitative relations form the basis of the subsequent analysis in this paper.

Coupling degree analysis is a method of determining the degree of dispersion between different economic systems and has been widely applied in various studies in fields related to economics and society. Generally, the calculation formula for the coupling degree is as follows:(9)$$C={\left[\frac{\prod_{i=1}^{n}U_{i}}{{\left(\frac{1}{n}\sum_{i=1}^{n}U_{i}\right)}^{n}}\right]}^{\frac{1}{n}}$$

In Equation (9), *C* represents the coupling degree between different dimensions, and *U_i_* represents the score of different systems. In the bidimensional state, Equation (9) is simplified as follows:(10)$$C={\left[\frac{U_{1}U_{2}}{{\left(\frac{U_{1}+U_{2}}{2}\right)}^{2}}\right]}^{\frac{1}{2}}=\frac{2\sqrt{U_{1}U_{2}}}{U1+U2}$$

Equation (10) is the main formula used in this paper to measure the coupling relationship between the quantity safety dimension and the input safety dimension.

## 4. Results

### 4.1. Descriptive Statistics

Table 2 shows the descriptive statistics of the indices. The first two columns are the maximum value and minimum value of the index and the mean and variance of the index, and the last column is the corresponding weight coefficient. Since the indicators in Table 1 are nondimensionalized, the values in column 1 and column 2 are all between 0 and 1, while the weight value of the last column is calculated by the entropy method, and the sum of the weights of all indicators is 1.

After dimensionless treatment, the negative indicators are treated as positive indicators. According to the results listed in the above table, the mean values of flood disaster area and drought disaster area in the negative index are both over 0.65, which indicates that the degrees of flood and drought disaster are relatively light. The weight values of the two are very low, at 0.9% and 1.4%, respectively, and have little impact on the overall index. The mean value of the import concentration degree and grain ration import adjustment degree of the grain market is approximately 0.4, and the weight values are 4.8% and 4.3%, respectively, which indicates that grain ration imports have insufficient positive effects on the food production security index. The mean value of pork consumption ratio variable was 0.182 with a weight of 9.4%, indicating a strong negative effect and a large weight. The mean value of the poultry consumption ratio variable is 0.757 with a weight of 1.1%. The negative impact and the weight are small. In conclusion, the negative effects of meat consumption growth, especially pork consumption growth, on food security cannot be ignored.

In the dimension of quantity safety, the production level index has a small value and a high weight, while the grain self-sufficiency index has a large value and a small weight. Because the difference in the mean value of the index is small, the grain yield per unit area and the grain per capita with higher weight play a more important role. In the quality safety dimension, the maximum qualified rate of food sampling is 0.738, but the weight is only 1.5%. The mean variables of food loss ratio and deeply processed food ratio are both less than 0.5, but their weights are 3.8% and 2.4%, which are significantly higher than the qualified rate of food sampling.

In the dimension of natural disasters, the weight of disaster management indices is generally higher than that of disaster degree indices. Overall, the damage caused by disasters is weak, but the effect of disaster prevention is very fruitful. Finally, the factor input dimension, the weight of water resource development and utilization, and labor input have values higher than 3.5%, and their mean values are over 0.4. The weights and the mean values are high, basically reflecting the important role of the two factors in food production. The weight of the capital index dropped to approximately 3% because of its high mean values and important input factor in food production. 

In the dimension of agri-food system transformation, except for the three negative indicators of pork and poultry and grain ration import, the other indicator with the highest weight is that of grain ration consumption, which reached 6.4%. Although the average values of grain ration consumption indicators are low at 0.32, the impact of ration grain security on food security cannot be ignored.

### 4.2. Food Security Index Analysis

Table 3 reports the measurement results of the food production security index. The first five columns are the scores of the food production security index and the five subdimensions. As described in the calculation method above, the safety index is obtained by the sum of the subdimensions. The last four columns in Table 3 report the proportions of different subdimensions in the production safety index, which reflects the relative importance of the subdimensions in the safety index.

First, the overall index of grain production more than doubled, from 0.298 in 2000 to 0.719 in 2019, indicating a marked improvement in the level of food production security. Before 2005, the fluctuation degree of the food production security index was high, and the increasing trend was not obvious. The index value of 2003 was even lower than that of 2000, basically reflecting the declining trend of food output from 1998 to 2003, and the index fluctuations of this period were caused mainly by the shortage in the total amount. After 2005, the production safety index showed an obvious trend of rapid growth, and the level of food production safety made great progress.

Second is the scores of different subdimensions. The change trend of the quantity security dimension is similar than that of the overall food security index. It fluctuates slightly before 2005, and the growth trend is not obvious. After 2005, it enters a stage of continuous growth, which continued until 2015. This process is known as the twelfth consecutive increase in grain. Compared with the low point in 2003, grain production capacity has greatly developed, increasing by 230 million tons. Although the absolute value of the quantitative security dimension increased rapidly, its proportion in the food security index remained stable, accounting for 15.4% in 2000 and 15% in 2019. From 2003 to 2015, the proportion of quantity security dimension increased steadily, which was highly coincident with the process of twelfth consecutive increase in grain. In 2015, the proportion reached the maximum 20.1%.

The score of quality safety dimension increased significantly, from 0.012 to 0.049, with an increase of more than 300%. Quality safety plays an increasingly important role in food security. The proportion of quality and safety scores increased from 4.2 percent in 2000 to 6.8 percent in 2019, with occasional fluctuations, but the overall trend was upward. Between 2000 and 2010, China’s food safety problems caused widespread concern. Tainted milk powder, gutter oil, Sudan red duck eggs, and other problematic food products seriously damaged the health of residents and caused adverse social impacts. The frequent occurrence of food safety problems has forced the Chinese government to improve the food safety supervision system, and quality safety is playing an increasingly important role in the food security.

Since the dimension of natural disasters includes two indices, disaster degree and disaster management, the score of this dimension integrates positive and negative effects. From the overall trend, the score of the natural disaster dimension is significantly lower than that of the factor input and agri-food system transformation dimensions, and is also slightly lower than that of the quantity security dimension. This is the dimension with the lowest score among all five dimensions, which reflects that natural disasters have a certain impact on food production security. From the perspective of the score of the dimension of natural disasters, the index increased continuously from 2000 to 2019, and even exceeded the dimension of quantity safety in some years, indicating that natural disaster management continued to improve and gradually became a key link in ensuring food production capacity. The proportion of the natural disaster score increased steadily from 5.5% in 2000 to 15.6% in 2019, which fully shows that the improvement of disaster response capacity plays a more important role in food security.

The factor input dimension score includes labor, land, capital, water, and energy. According to the trend in the score change, the factor input subdimension can be divided into two stages: before 2007 and after 2007. Before 2007, the dimension score was approximately 0.1, but after 2007, it gradually increased to more than 0.15, and even approached 0.25 in some years. From the perspective of factor input dimension, the proportion data maintained increasing momentum before 2015. After 2015, the growth trend of the factor input score fell, partly due to the decreasing returns of capital factors. The same amount of capital input increase could not be exchanged for the continuous and equal increase of output. In addition, environmental problems caused by the excessive application of chemical fertilizers and pesticides gradually emerged. The central government began to advocate for soil testing and formula fertilization, which resulted in zero growth of pesticides and chemical fertilizers, and the impact of inputs of relevant factors on food security declined.

The score of the dimension of agri-food system transformation fluctuated greatly. After reaching the high point of 1.81 in 2007, it began to decline, and then decreased to 0.09 in 2013. After that, it began to increase again, and increased to 0.22 in 2019. In terms of the proportion of agri-food system score, the trend of decline was obvious, from 59.9% in 2000 to 30.4% in 2019, indicating that the positive effect of agri-food system transformation on food security declined significantly. This is mainly due to the growth of meat consumption, such as pork and poultry, which has increased the demand for feed grains and increased the import of feed grains. The transformation of the agri-food system has caused great pressure on the balance of food supply and demand, and its contribution to food security is gradually declining.

Finally, the coupling relationship between agri-food system transformation and food quantity security is discussed after the scores of different sub-dimensions are calculated by entropy method. According to Figure 2, the coupling degree of agri-food system transformation and the quantity security experienced a process of first rising and then falling. The index of the coupling degree between 2003–2014 has shown long-term growth, increasing from 0.71 to close to 1, then gradually decreasing to 0.94 in 2019, and agri-food system transformation and quantity security coordination is falling. 

According to the contents of the food security index system, the food quantity security was decomposed into food production and food self-sufficiency, and the coupling degree between the two secondary indices and the transformation dimension of the agri-food system was calculated. According to Figure 3, the coupling degree between grain production capacity and agri-food system transformation reached a peak of 0.99 in 2015, and then gradually decreased to 0.9 in 2019, which was mainly influenced by the twelve years of consecutive increase in grain output in 2015 followed by and then the insufficient potential of subsequent grain increase. However, the coupling degree between the dimension of grain self-sufficiency and the dimension of agri-food system transformation did not increase significantly from 2000 to 2019. On the contrary, after 2013, the index showed an obvious downward trend, decreasing from 0.87 in 2013 to 0.6 in 2019. The reason is that the demand for meat and feed increases synchronously due to the transformation of the agri-food system. However, China’s production capacity of feed grains, especially the production capacity of soybeans, is insufficient, which leads to a decline in the feed grain self-sufficiency rate, and the transformation of the agri-food system is diverging from the guarantee of food self-sufficiency rate.

## 5. Discussion and Conclusions

Based on a detailed review of the agri-food system transformation both globally and in China, this paper constructs an analytical framework of the impact of agri-food system transformation on food security and discusses the food security problems in China after 2000. Firstly, the food security level from 2000 to 2019 was calculated by establishing a food security index system that includes data from five dimensions: food quantity security, food quality security, factor input guarantee, natural disaster impact, and agri-food system transformation. On this basis, the coupling adaptation relationship between agri-food system transformation and food quantity security was further analyzed. The results show that the proportion of the dimension of agri-food system transformation in the food security index has a significant downward trend, and the overall positive effect on food security is decreasing. At the same time, the coupling degree between the transformation dimension of agri-food system and the level of grain production has a downward trend due to the decrease of grain increase potential. In addition, due to the growth of meat demand in the transformation process of the agri-food system, the demand for feed grains increased, and a large number of feed grains such as soybeans and corn were imported. The self-sufficiency of grain declined, and the agri-food system gradually deviated from the food security system.

In the near future, the grain production quantity and the supply and demand structure still face challenges. How to fully utilize the potential of agricultural production and ensure national food security remains a very important issue. According to the results of this paper, improving food production capacity on the supply side is not the only way to maintain food security. Optimizing food consumption structure on the demand side is also an important means to improve food security. Specifically, groups should advocate for the consumption of poultry rather than pork. Poultry meat is rich in unsaturated fatty acids, and long-term consumption can significantly reduce the incidence of various chronic diseases and malignant tumors [27,28]. At the same time, poultry production has a more significant effect on reducing feed consumption than pork production. From the policy level, it is necessary to strengthen the publicity of scientific knowledge on food and nutrition. Guiding urban and rural residents to increase poultry consumption is a feasible way to improve nutrition and health of residents and enhance food security. Second, food waste should be curbed and food utilization improved. China wastes about one-sixth of its total grain output in production each year. Meanwhile, the waste of outdoor dining and dumping of expired food is also considerable [29]. Therefore, advocating for a scientific and reasonable concept of food consumption and curbing table waste is also an important means to ensure food security from the demand side. Third, regarding natural disasters and extreme climate issues, the Chinese government has taken measures, such as building high-standard farmland and improving irrigation and water conservancy facilities to address risks. In the nearly future, developing early warning systems and establishing a normalized rapid response mechanism will play a crucial role. From a global perspective, many developing countries are experiencing or will experience the transformation of agri-food system represented by the increase of meat consumption preference. China’s situation in the face of agri-food system transformation and corresponding solutions will provide reference for other developing countries. As a country with a large population and a large grain supply, maintaining a relatively high grain self-sufficiency rate for a long period of time will significantly reduce international food demand, calm international food prices, and help better achieve the policy goals of fighting hunger and poverty.

There are still some shortcomings in the paper. The connotation of the transformation of the agri-food system is not limited to the structural adjustment of food production and distribution. According to the definition of the agri-food system, other elements and exchange relationships in the value chain of agriculture are also part of the transformation. For example, the increase of meat consumption leads to the increase of carbon emissions, the massive use of fertilizers leads to non-point source pollution, and the increase of meat consumption ratio leads to the nutrition problems for the residents, which can also be included in the research on the transformation of agri-food system. In future studies, multiple factors can be further considered to deepen the research framework of the agri-food system.

## Figures and Tables

**Figure 1 foods-12-02906-f001:**
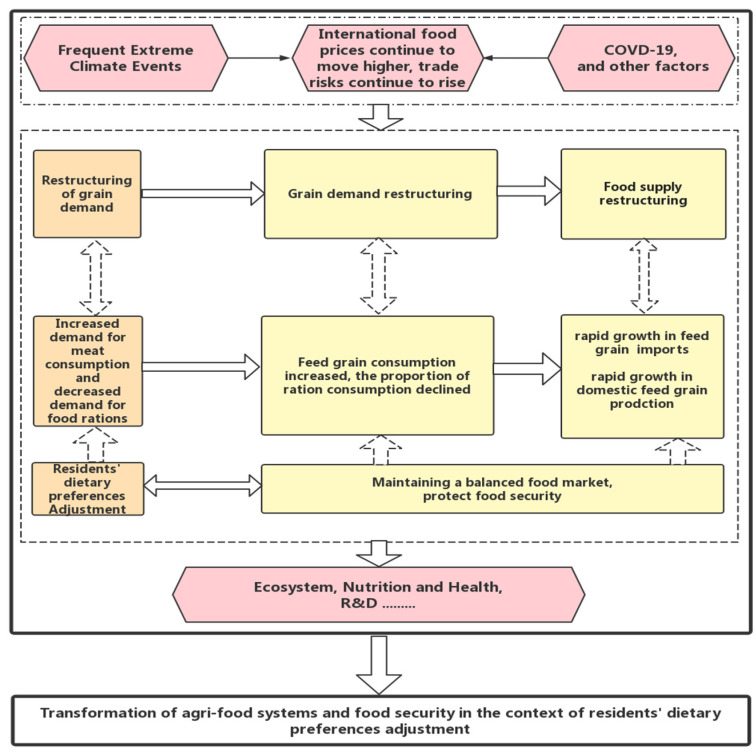
The logical between the transformation of agri-food system and food security.

**Figure 2 foods-12-02906-f002:**
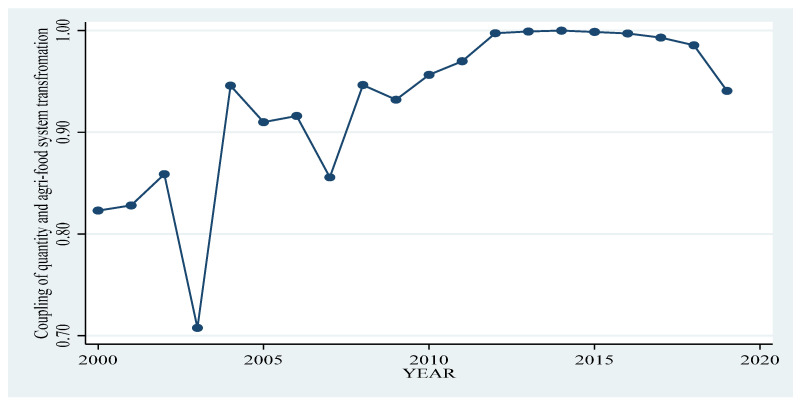
Coupling trend of quantity security dimension and agri-food system transformation.

**Figure 3 foods-12-02906-f003:**
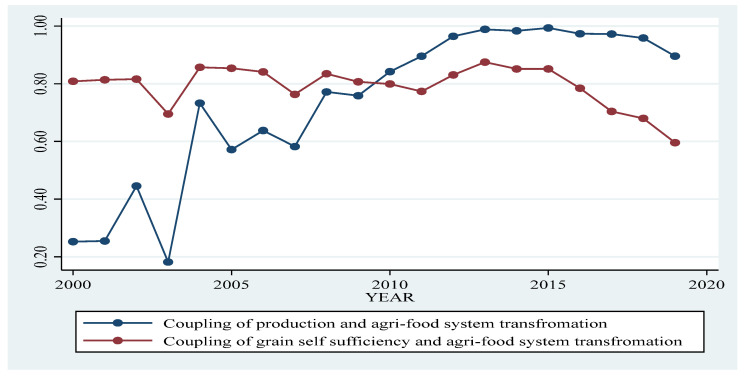
The trend of coupling degree between grain production level, self-sufficiency and agri-food system transformation.

**Table 1 foods-12-02906-t001:** Food security index system.

First-Level Indicators	Second-Level Indicators	No.	Third-Level Indicators
Quantity safety dimension	Grain production levels	1	Grain output per unit area
		2	Grain per capita
	Self-sufficiency in grain	3	Grain self-sufficiency rate
		4	Grain ration self-sufficiency rate
		5	Level of grain reserves
Quality safety dimension	Quality level	6	Food sampling inspection qualified rate
		7	Food loss ratio
		8	Proportion of highly processed grain
Natural disaster dimension	Extent of disaster	9	Areas affected by floods
		10	Drought-stricken areas
	Disaster management	11	Flood relief rate
		12	Drought relief rate
		13	Soil erosion control area
Factor input dimension	Development and utilization of water resources	14	Effective irrigated area
		15	Reservoir capacity
	Land investment	16	Proportion of grain sown area to total sown area
	Input of labor force	17	Labor force in primary industry
	Capital investment	18	Agricultural machinery power station
		19	Amount of fertilizer used
	Energy consumption	20	Rural electricity consumption
		21	Number of rural hydropower stations
Agri-food system transformation	Adjustment of grain consumption structure	22	Proportion of grain ration
		23	Proportion of feed grain
		24	Ratio of meat consumption to food consumption
	Meat consumption structure adjustment	25	Proportion of pork consumption
		26	Proportion of beef and mutton consumption
		27	Proportion of poultry consumption
	Adjustment of grain import structure	28	Grain ration import adjustment degree
		29	Share of grain imports
		30	Market concentration of grain imports

**Table 2 foods-12-02906-t002:** Descriptive statistics of the indices.

	(N = 20)	(N = 20)	(N = 20)
	Min Max	Mean (Std)	w_j
Grain output per unit area	0.000, 1.000	0.436 (0.357)	0.052
Grain per capita	0.000, 1.000	0.547 (0.358)	0.034
Grain self-sufficiency rate	0.000, 1.000	0.520 (0.330)	0.032
Grain ration self-sufficiency rate	0.000, 1.000	0.585 (0.244)	0.014
Level of grain reserves	0.000, 1.000	0.552 (0.230)	0.014
Food sampling inspection qualified rate	0.000, 1.000	0.738 (0.294)	0.015
Food loss ratio	0.000, 1.000	0.300 (0.239)	0.038
Proportion of highly processed grain	0.000, 1.000	0.471 (0.270)	0.024
Areas affected by floods	0.000, 1.000	0.793 (0.235)	0.009
Drought-stricken areas	0.000, 1.000	0.699 (0.273)	0.014
Flood relief rate	0.000, 1.000	0.434 (0.304)	0.034
Drought relief rate	0.000, 1.000	0.466 (0.299)	0.031
Soil erosion control area	0.000, 1.000	0.424 (0.280)	0.032
Effective irrigated area	0.000, 1.000	0.437 (0.365)	0.053
Reservoir capacity	0.000, 1.000	0.507 (0.369)	0.041
Proportion of grain sown area to total sown area	0.000, 1.000	0.675 (0.289)	0.016
Labor force in primary industry	0.000, 1.000	0.502 (0.355)	0.038
Agricultural machinery power station	0.000, 1.000	0.542 (0.330)	0.030
Amount of fertilizer used	0.000, 1.000	0.593 (0.342)	0.028
Rural electricity consumption	0.000, 1.000	0.555 (0.354)	0.033
Number of rural hydropower stations	0.000, 1.000	0.580 (0.445)	0.052
Proportion of grain ration	0.000, 1.000	0.322 (0.316)	0.064
Proportion of feed grain	0.000, 1.000	0.531 (0.305)	0.026
Ratio of meat consumption to food consumption	0.000, 1.000	0.515(0.253)	0.019
Proportion of pork consumption	0.000, 1.000	0.182 (0.246)	0.094
Proportion of poultry consumption	0.000, 1.000	0.757 (0.261)	0.011
Proportion of beef and mutton consumption	0.000, 1.000	0.422 (0.263)	0.028
Grain ration import adjustment degree	0.000, 1.000	0.400 (0.317)	0.048
Share of grain imports	0.000, 1.000	0.542 (0.330)	0.029
Market concentration of grain imports	0.000, 1.000	0.409 (0.305)	0.043

**Table 3 foods-12-02906-t003:** Food security index and proportion of different dimensions.

Year	Food Security Index	Score					%				
		Quantity Safety	Quality Safety	Natural Disaster	Factor Input	Agri-Food System Transformation	Quantity Safety	Quality Safety	Natural Disasters	Factor Inputs	Agri-Food System Transformation
2000	0.298	0.046	0.012	0.016	0.057	0.167	0.154	0.042	0.055	0.191	0.559
2001	0.272	0.041	0.007	0.022	0.058	0.144	0.149	0.025	0.081	0.214	0.531
2002	0.330	0.046	0.054	0.028	0.060	0.142	0.139	0.164	0.086	0.182	0.430
2003	0.287	0.027	0.025	0.022	0.055	0.158	0.095	0.088	0.076	0.191	0.550
2004	0.307	0.055	0.020	0.060	0.063	0.109	0.181	0.066	0.195	0.205	0.354
2005	0.312	0.050	0.022	0.050	0.070	0.120	0.159	0.070	0.161	0.225	0.385
2006	0.351	0.059	0.032	0.037	0.085	0.137	0.167	0.092	0.106	0.243	0.391
2007	0.442	0.057	0.048	0.056	0.100	0.181	0.130	0.109	0.127	0.225	0.408
2008	0.461	0.074	0.031	0.053	0.159	0.144	0.160	0.068	0.115	0.344	0.313
2009	0.494	0.066	0.034	0.081	0.171	0.141	0.134	0.069	0.164	0.346	0.286
2010	0.475	0.070	0.034	0.062	0.182	0.127	0.147	0.071	0.131	0.383	0.268
2011	0.527	0.080	0.031	0.093	0.192	0.131	0.151	0.060	0.176	0.365	0.249
2012	0.526	0.090	0.044	0.076	0.213	0.104	0.171	0.083	0.145	0.404	0.197
2013	0.529	0.098	0.047	0.070	0.223	0.090	0.185	0.090	0.133	0.422	0.171
2014	0.537	0.100	0.037	0.068	0.231	0.100	0.187	0.069	0.127	0.431	0.186
2015	0.540	0.108	0.034	0.059	0.240	0.098	0.201	0.063	0.109	0.445	0.181
2016	0.564	0.104	0.033	0.066	0.240	0.121	0.185	0.058	0.116	0.426	0.215
2017	0.591	0.103	0.036	0.080	0.243	0.130	0.174	0.061	0.135	0.410	0.220
2018	0.608	0.101	0.046	0.083	0.236	0.142	0.166	0.075	0.137	0.387	0.234
2019	0.719	0.108	0.049	0.112	0.232	0.218	0.150	0.068	0.156	0.323	0.304

## Data Availability

The data presented in this study are available on request from the corresponding author.

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
