# Peer review of "Agricultural Food System Transformation on China’s Food Security"

_foods, 2023, doi:10.3390/foods12152906_

Round 1
Reviewer 1 Report (Previous Reviewer 4)
Dear Dr. Meta Wang,
The manuscript entitled “Agricultural food system transformation on China's food security" needs major revision.
The discussion presupposes a comparison of the own results with the works available in this direction. However, in section “Discussion and Conclusion” there are no references on any papers at all and statements written there are not proved. For example:
Lines 442-444. “Poultry meat is rich in unsaturated fatty acids, and long-term consumption can significantly reduce the incidence of various chronic diseases and malignant tumors”.
This statement is not shown in the article and is not confirmed by references to literature sources.
The article also lacks clear conclusions and recommendations.
Yours sincerely,
Author Response
Responses to Reviewer1:
Thank you for your comments concerning our manuscript entitled “Food Security in Developing Countries under Agri-food System Transformation: Analysis Based on Chinese Data” (Manuscript ID: foods-2515077). Those comments are all valuable and very helpful for revising and improving our paper. Revisions are marked up using the “Track Changes” function, the edited words/sentences are highlighted in the revised manuscript and the response to the comments are as following.
Comment 1
The discussion presupposes a comparison of the own results with the works available in this direction. However, in section “Discussion and Conclusion” there are no references on any papers at all and statements written there are not proved.
For example: Lines 442-444. “Poultry meat is rich in unsaturated fatty acids, and long-term consumption can significantly reduce the incidence of various chronic diseases and malignant tumors”. This statement is not shown in the article and is not confirmed by references to literature sources.
Reply: Thanks for your comments. In the revised version, we have added citations [27],[28],[29] in lines 456-467.The revised part has been highlighted in yellow.

Reviewer 2 Report (Previous Reviewer 2)
My apologies.... there are some serious deficiencies within this paper....
The title does not reflect your intentions - you are talking about impact - but you have also restricted your analysis to only two commodities: chicken and grain. You cant talk about meat as you dont look at beef/sheep/goat and its apparent that while pork production in China is intensive, you dont discuss this either
Figure 1 is not logical. What's changed is the consumer demand for meat and that impacts the demand for grain > implications for both domestic production and imports
Your references are woefully inadequate... if youre going to talk food security why do you not refer to the FAO or IFPRI? Both of these organisations work extensively in this space and have the best data.
Why do you not use the four dimensions of food security: availability, accessibility, utilisation and stability. These are already established. In more recent times, the HLPE have revised food security to include two new dimension: sustainability and agency... no mention at all of these....
Food systems... again.... why do you not utilise the FAO? Their information is not only publically available but their definitions are widely used by most other people in this space [including myself]
I began by reviewing your English, but by the end of the first page I gave up... the grammar is wrong, the tense is wrong and on far too many occasions you use inappropriate words - like ration - for example. This implies that grain consumption in China is rationed - and I doubt very much if that is the case
Why develop a new food security index? There are already two such indexes already in use: the FAO FSI and the Economist Global FSI. The measures utilised to develop these indexes are well developed and publically available. No mention is made that these even exist.....
Being aware now that these indexes do exist, where are the deficiencies? Where and how can you improve these measures???
Use this study then as a case study to compare and contrast your data and approach with that utilised by FAO and the Economist and potentially you have something worthy of publication
as above
Author Response
Responses to Reviewer2:
Thank you for your comments concerning our manuscript entitled “Food Security in Developing Countries under Agri-food System Transformation: Analysis Based on Chinese Data” (Manuscript ID: foods-2515077). Those comments are all valuable and very helpful for revising and improving our paper. We have studied comments carefully and have made revisions which we hope meet with your requirements. Additionally, the revised manuscript was edited for proper English language, grammar, punctuation, spelling, and overall style by one or more of the highly qualified native English-speaking editors at AJE. Revisions are marked up using the “Track Changes” function, the edited words/sentences are highlighted in the revised manuscript and the response to the comments are as following.
Reviewer 2
My apologies.... there are some serious deficiencies within this paper....
Comment 1
The title does not reflect your intentions - you are talking about impact - but you have also restricted your analysis to only two commodities: chicken and grain. You can’t talk about meat as you don’t look at beef/sheep/goat and its apparent that while pork production in China is intensive, you don’t discuss this either.
Reply: Thanks very much for your valuable advice.
The connotation of agri-food system transformation is abundant. Our article focuses on the impact that food demand structural change on food security. And the structural change is mainly about the increase in meat consumption and the decrease in grain ration consumption. We cannot follow up all the topics of agri-food systems transformation in our article (we have stated these shortcomings at the end of the manuscript), the article only mentions the core issue of the transformation process.
Besides, pork/poultry/beef/mutton consumption are all included in the indicator system in the article. Please refer to the second-level indicator “Meat consumption structure adjustment” for details. Particularly, we pay more attention to pork and poultry in this article. This is mainly because the production of these two kinds of meat requires a large amount of feed grain. Overall, we believe that the title is a reasonable choice.
Comment 2
Figure 1 is not logical. What's changed is the consumer demand for meat and that impacts the demand for grain > implications for both domestic production and imports.
Reply:Thank you for the comments. We have revised Figure1 as your suggestions to make it more logical.
Comment 3
Your references are woefully inadequate... if you’re going to talk food security why do you not refer to the FAO or IFPRI? Both of these organisations work extensively in this space and have the best data.
Reply: Sorry for the lack of references. We have added several refences from FAO in the revised manuscript to make it more convincing, such as refence [25][26]. Also, we take the FAOSTAT database as the major source of data citation, as well as the OECD-FAO Agricultural Outlook, please refer to the reference [4][6] for more details.
Comment 6
I began by reviewing your English, but by the end of the first page I gave up... the grammar is wrong, the tense is wrong and on far too many occasions you use inappropriate words - like ration - for example. This implies that grain consumption in China is rationed - and I doubt very much if that is the case.
Reply: Sorry for the wrong inappropriate words. We have replaced “ration” to “grain ration” in revised manuscript. We use the word “grain ration” to refer to the food that people eat. Additionally, the revised manuscript was edited for proper English language, grammar, punctuation, spelling, and overall style by one or more of the highly qualified native English-speaking editors at AJE.
Comments 4/5/7
Comment 4
Why do you not use the four dimensions of food security: availability, accessibility, utilisation and stability. These are already established. In more recent times, the HLPE have revised food security to include two new dimension: sustainability and agency... no mention at all of these....
Comment 5
Food systems... again.... why do you not utilise the FAO? Their information is not only publically available but their definitions are widely used by most other people in this space [including myself]
Comment 7
Why develop a new food security index? There are already two such indexes already in use: the FAO FSI and the Economist Global FSI. The measures utilised to develop these indexes are well developed and publicly available. No mention is made that these even exist.....
Being aware now that these indexes do exist, where are the deficiencies? Where and how can you improve these measures???
Use this study then as a case study to compare and contrast your data and approach with that utilised by FAO and the Economist and potentially you have something worthy of publication.
Reply: Thank for the elaborate comments about our food security index system. We have added a part to explain our motivation for establishing a food security system in the beginning of “Methodology”. The addition part has been highlighted in yellow.
We mentioned the FAO FSI and the Economist FSI in the revied 3. Methodology. As you stated, FAO and the Economist are all have developed widely accepted food security index systems. But these index systems may be not suitable for our research and the purpose of establishing a new FSI in this article is to study the impact of agri-food system transformation on food security. The FSI is a means rather than our research object.

Reviewer 3 Report (New Reviewer)
The ms. “Agricultural food system transformation on China's food security” (Ms. Ref. No. foods-2515077-v1) presents the evolution of the food agricultural system and market in China from the viewpoint of the food security, approached through various specific descriptors such as the food security index, quantity/quality scores, agrifood system transformation score, natural disaster score etc. It looks like the manuscript has undergone previous revision.
The investigated topic falls within the aims and scopes of the Foods journal.
There is a lot of work involved and the ms. has evident merit and potential to get published.
However, there are some points to be addressed to improve the quality of the review.
Major issues:
11.. The aim of the study has not been stated in the Introduction. Usually, this should be provided at the end of the section, stating the aim/approach, and briefly describing the methodology.
22. In the Introduction: The authors should point out the originality of their work (in terms of topic/approach/methodology) compared to the current literature landscape in the field. Please elaborate on the novelty/originality.
33. I liked the manuscript, and the investigated topic and approach were very interesting to me. However, it was surprising that the issue of food fraud was not even mentioned. Still, it is a serious matter and acknowledged by the authorities worldwide, not only in China. In my opinion, a sub-section should be dedicated to food fraud and counterfeited products as a part of the food security issue. In this context, it is worth pointing out that the fraud techniques have become in some cases more refined; on the other hand, the methods of detection have evolved. However, there are cases in which the proposed methods fail, as was recently demonstrated by Hanganu et al. (When detection of food fraud fails). In this respect, regarding milk fat, which is probably the most susceptible to worldwide adulteration food commodity, it was shown that even methods based on cutting-edge techniques such as the 1H-NMR spectroscopy may fail in detecting the addition of non-dairy fats. The authors have even suggested “clever” mixtures of fats and oils of non-dairy origin that would perfectly imitate the behavior and the 1H-NMR spectra of genuine dairy products. In addition, also in the case of milk and dairy sector (expensive foods and limited production compared to the market demand), there are other examples of limitations of the currently existing methods for testing their authenticity. For example, it is worth pointing out that it was recently suggested (Ivanova et al., Foods 2022, 11(10), 1466) that the saponification value of fats and oils computed from the 1H-NMR spectra may be an useful tool in detecting adulteration of milk and dairy products though replacement or addition of other non-dairy fats. However, a foreseen limitation is that an altered milk fat composition would be difficult to detect through the saponification value criterium if coconut oil is used as an adulterant, given its high content in medium chain length fatty acids. Please revise and elaborate on this issue in the Discussion section.
44. Please revise English and typos.
55. The plural for index is indices. Please update.
Given the completed score sheet and the comments above, after careful evaluation, the ms. “Agricultural food system transformation on China's food security” (Ms. Ref. No. foods-2515077-v1) needs Major Revision according to comments.
Minor English editing, in my opinion.
Author Response
Responses to Reviewer 3:
We feel great thanks for your professional review work on our manuscript. We have read your suggestions with a serious attitude and adopted them. These comments are all valuable and helpful for improving our manuscript. The revised manuscript was edited for proper English language, grammar, punctuation, spelling, and overall style by one or more of the highly qualified native English-speaking editors at AJE. Revisions are marked up using the “Track Changes” function, the edited words/sentences are highlighted in the revised manuscript and the response to the comments are as following.
Reviewer3
Comment 1
The aim of the study has not been stated in the Introduction. Usually, this should be provided at the end of the section, stating the aim/approach, and briefly describing the methodology.
Reply: Thanks for your comments. Based on your suggestion, we have added research objectives at the end of “introduction”. The revised part has been highlighted in yellow from lines 58-61.
Comment 2
In the Introduction: The authors should point out the originality of their work (in terms of topic/approach/methodology) compared to the current literature landscape in the field. Please elaborate on the novelty/originality.
Reply: We emphasized the innovation of our research at the end of “introduction” in the revised manuscript from line 59-60. We also elaborate the novelty of out manuscript at the end of the “Literature Review”. The revised part has been highlighted in yellow.
Comment 3
However, it was surprising that the issue of food fraud was not even mentioned. .............................. However, there are cases in which the proposed methods fail, as was recently demonstrated by Hanganu et al. (When detection of food fraud fails). In this respect, regarding milk fat, ..................For example, it is worth pointing out that it was recently suggested (Ivanova et al., Foods 2022, 11(10), 1466)........................Please revise and elaborate on this issue in the Discussion section.
Reply: Thank for the suggestion. The issue of food fraud does threaten the global food security. We added a part of discussion about food fraud in “Literature Review”, and the revised part has been highlighted in yellow. As your suggestion, Ivanova et al (2022) and Hanganu &Chira (2021) have already been referenced.
Comment 4
Please revise English and typos.
Reply: We have carefully proofread the revised version and corrected all typos. Meanwhile, a professional polishing agency was also found to edit the language of the manuscript.
Comment 5
The plural for index is indices. Please update.
Reply: Thank you for pointing out that detail. we have changed the plural of “index” into “indices”,and the revised part has been highlighted in yellow.

Round 2
Reviewer 1 Report (Previous Reviewer 4)
The revised manuscript can be published.
Author Response
Responses to Reviewer1:
Thanks again for your comments concerning our manuscript entitled “Food Security in Developing Countries under Agri-food System Transformation: Analysis Based on Chinese Data” (Manuscript ID: foods-2515077). Those comments are all valuable and very helpful for revising and improving our paper.
Wishing you all the best
Sicheng Zhao, Tingyu Li, Guogang Wang *
2023-07-25
Reviewer 2 Report (Previous Reviewer 2)
I asked for a major review.... that requires a rewrite - not a few extra paragraphs.
Since you choose not to define food security in the way it is more broadly accepted (FAO 2006), I suggest you reposition this paper completely. Lets call this a grain self sufficiency index.... Developing this in the way that you do will cover both the changes in supply and demand, inputs, trade and policy, disaster management and mitigation
Forget about food systems transformation.... thats inevitable and indeed is a consequence of both changing demand and the impact of government policy... and thats an area which is currently weak: the governmnet now recognise that China CANNOT produce enough grain and hence it must import... it also demonstrates the investments govt has made in transport and infrastructure to reduce food loss [not waste] - in other words - as you dont look at food utilisation how can you address food waste?
The other issue here relates to urbanisation and its implications.... consumers dont buy grain [except rice] - they buy flour/processed food products derived from that grain... as hence food manufacturing becomes as issue and with that, issues such as energy [beyond just fuel]
Its finally on P5 that you define food security - as you choose to do so in a completely different way - this is where your key problems begin - and why I referred you to the FAO FSI and the Economists Global FSI. Youre completely right - they dont meet you needs - but you dont measure/evaluate food security either - so if we look at grain self sufficiency - you can probably position this paper better
You even say this yourself [see P5 L181-185]
So what Im suggesting here is a MAJOR rewrite/repositioning of the paper that will accommodate the modelling you have undertaken. You also need to rewrite your discussion and conclusions to address the findings and align these with government policy and highlight areas where intervention is required
Disaster prevention is of interest to me [and I suspect to other readers] - so what does and how does the government of China mitigate this??? Climate change is real - and with that the frequency of adverse weather events - and that is putting pressure on our global food systems
Weak..... a final copy of the revised manuscript must be edited before submission
Author Response
Responses to Reviewer2:
Thanks again for your comments concerning our manuscript entitled “Food Security in Developing Countries under Agri-food System Transformation: Analysis Based on Chinese Data” (Manuscript ID: foods-2515077). Those comments are all valuable and very helpful for revising and improving our paper. The edited words/sentences are highlighted in the revised manuscript and the response to the comments are as following.
Comments 1
I asked for a major review.... that requires a rewrite - not a few extra paragraphs......................
Since you choose not to define food security in the way it is more broadly accepted (FAO 2006), I suggest you reposition this paper completely. ...................................
So what Im suggesting here is a MAJOR rewrite/repositioning of the paper that will accommodate the modelling you have undertaken. You also need to rewrite your discussion and conclusions to address the findings and align these with government policy and highlight areas where intervention is required ....................................
Reply:
We are strongly agree with your academic viewpoints, but rewriting this article may not be a rational choice.
Based on your comment, we are now on the same page.You also agree that calculating the FSI (Food Security Index) is not the focus of this article. The impact of the the agri-food system transformation on food security is more improtant.There are several reasons that we dont rewrite the thesis.
Firstly ,“ grain self sufficiency index” may not include the meat consumption structure change and the food quality stafety . This definition of food security may be too narrow and limmitted.
Secondly, The issues of urbanization and biomass production are far beyond our discussion.Adding these factors may require writing a new article.
Comments 2
Disaster prevention is of interest to me [and I suspect to other readers] - so what does and how does the government of China mitigate this??? Climate change is real - and with that the frequency of adverse weather events - and that is putting pressure on our global food systems.
Reply:
Based on your suggestion, we have add a part in “Discussion and Conclusion” to
Show China's efforts and future priorities in addressing climate change.The modified part has been highlighted in yellow.
Comments 3
Comments on the Quality of English Language: Weak..... a final copy of the revised manuscript must be edited before submission.
Reply:
This article has sought a professional polishing agency for polishing.The attachment is a proof of polishing for this article. We will once again request the organization to improve the language of the article before submitting the final version
Wishing you all the best
Sicheng Zhao, Tingyu Li, Guogang Wang *
2023-07-25

Reviewer 3 Report (New Reviewer)
The revised ms. “Agricultural food system transformation on China's food security” (Ms. Ref. No. foods-2515077-v2) was significantly improved compared to its initial version. The authors have addressed all the reviewers’ concerns in an appropriate manner. From the scientific viewpoint, I do not have further comments. I still recommend English revision, preferably by a professional editing service.
I recommend English revision, preferably by a professional editing service.
Author Response
Responses to Reviewer3:
Thanks again for your comments concerning our manuscript entitled “Food Security in Developing Countries under Agri-food System Transformation: Analysis Based on Chinese Data” (Manuscript ID: foods-2515077). Those comments are all valuable and very helpful for revising and improving our paper.
This article has sought a professional polishing agency for polishing.The attachment is a proof of polishing for this article. We will once again request the organization to improve the language of the article before submitting the final version
Wishing you all the best
Sicheng Zhao, Tingyu Li, Guogang Wang *
2023-07-25
This manuscript is a resubmission of an earlier submission. The following is a list of the peer review reports and author responses from that submission.
Round 1
Reviewer 1 Report
- I recognize the scientific value and originality of the contribution, but the paper needs deep and major revisions. Be careful; the references are missing for the first two pages of the paper.
Many sentences, ideas, and paragraphs are without references.
The quality of the English is very, poor. I am afraid that it is often quite difficult to understand clearly and without ambiguity what the authors mean. Many sentences are unclear. paper missed flows between sentences and paragraphs.
I recommend extensive editing of the English language and style using a professional proofreading service.
Reviewer 2 Report
Good effort....
However, the paper would benefit from a significant restructuring
Your Introduction should be exactly that - an introduction - I dont want to see figures and tables in the Intro - rather these should appear in the following literature review. The best way to think about the Intro is that it is a summary of the lit review - it should follow the same funnel approach from the macro [global] issues to the more specific micro issues - and conclude with your aims and objectives.
To the lit review itself.... start with food security - and within that space - talk about global food production [supply] and demand. Talk about the various externalities impacting supply and the demand - and its here that you might now move to talk about meat. That should then lead you into the discussion around feed grains and the increasing global trade. Now talk about China - and how the demand for meat is growing - the demand for grain is growing - local production and imports. Personally, I see little need to embark on any serious discussion or review of food systems - food security is your focus
Given that your objective is the develop a tool to evaluate a food security index - some review of other authors attempts to do the same is appropriate and required - with the focus here on the positive/negative aspects of these tools/models. Currently this section is missing....
Now we move to the methodology whereupon you and your co-authors develop the model
Results then a straight forward - having developed the model - from the database [presumably the Chinese govt statistics office] - we plug in the data and describe what it shows
Your discussion is primarily two parts: analyse the model and indeed refine it and/or note any limitations - and discuss the implications
The conclusion also is quite straight forward - the model showed that... Its here that you suggest that other researchers use your model in their countries
Your English comprehension is generally good, but I always recommend that you have the paper read by an experienced english editor prior to submission
Reviewer 3 Report
The paper addresses the very important issue of food security in China. This topic is all the more important given that it is the country with the largest population in the world. I do, however, have some serious comments on the content, as well as on the language or editing of the text. I include them below.
Section 1 and subsection 3.2 are missing references. They should be completed, although it will unfortunately require renumbering other further references.
The last sentence of Section 2 should be converted into an explicit goal of the research.
Remove the yellow highlights in Table 2 or indicate clearly in the text what they mean.
Why the different words are used in the names of the second dimension in Table 1 and Table 3, i.e., “quality” and “structural”) ?
The word “security” should be used in the names of the first two dimensions in Table 1 and Table 3, not “safety”.
I also believe that sentences are often too long, thus becoming incomprehensible or even illogical. Therefore, the text should be checked and corrected by someone with a good knowledge of English.
15-16 consumption… consumption. > and feed consumption.
34 FAO > Food and Agriculture Organization of the United Nations (FAO)
35-36 On the other… impact of > This is influenced by exogenous factors as
42 tons, energy > tons. It is estimated that energy
44 In terms of meat consumption, global > The global
46-49 In terms of total consumption, East Asia, the Americas and Europe have the highest level of meat consumption, reaching 100 million tons, 90 million tons and 60 million tons respectively in 2019. > In terms of total consumption in 2019, East Asia, the Americas and Europe have the highest level of meat consumption, reaching 100 million tons, 90 million tons and 60 million tons, respectively.
49 The three > These three
55 average. The local > average. However, the local
62-67 China… value. (This sentence is too long and incomprehensible. It should be reworded and also divided into two separate sentences.)
103-108 External… serious. (This sentence also should be reworded and divided.)
101-103 Figure 2… structure. (Move this sentence before Figure 2.)
148 mortality[11] > mortality [11]
149 Agricultural > agricultural
202 agricultural food > agri-food
211 agricultural food > agri-food
212 dimension. > dimension (Table 1). (I propose to refer to Table 1 at this place. Also, since tables should not be divided into pages, I propose to place Table 1 right here and start the paragraph after it with the sentence “The quantity security...”.)
224-225 The specific… table. (Remove this sentence.)
227 (The title of Table 1 is incomprehensible, reword it.)
251 i, j > i, j (Write i and j in italic.)
265 n > n (Write n in italic.)
273 i, m > i, m (Write i and m in italic.)
276 i > i (Write i in italic.)
276 mk > mk (Write mk in italic, k in the subscript.)
287 C > C (Write C in italic.)
288 Ui > Ui (Write Ui in italic, i in the subscript.)
293 4. Results (Move the title of this section to the next page.)
302 (Make a one line space below Table 2.)
303 above table > Table 2
308-309 4.3% respectively > 4.3%, respectively
324 2.4% , which > 2.4%, which
337 4.8% respectively > 4.8%, respectively
338 0.4% respectively > 0.4, respectively
340 5.2. Food > 4.2. Food
347 (Move the title and Table 3 to the next page.)
414 (Make a one line space between text and Figure 3.)
418 figure 3 > Figure 3
418-422 According… falling. (This sentence is too long and incomprehensible. Reword it. I also propose to move this fragment, i.e. “Finally… falling.” before Figure 3, because tables and figures should be referred to before they appear in the paper, not after.)
428 12th > 12 years old
439-440 The names of the trends in Figure 4 do not correspond to the title of this figure.
473 1700-18 (This clause is incomprehensible and should be corrected.)
493 residents can > residents, which can
501-542 (The way references are described should be changed. They do not meet the requirements of the Journal.)
Reviewer 4 Report
The article can be useful for creating an analytical framework of the impact of the transformation of the agro-food system on China's food security.
In my opinion, the article is not suitable for publication in the journal “Foods”, which is devoted to research in the framework of “Science of Food”.
Authors should submit the article to a journal with a topic that corresponds to the direction of the research. I would like to ask the authors to pay attention to the following remark: there is no conflict between Ukraine and Russia. There is a military invasion of russian troops on the territory of the independent boundary country Ukraine.